# Considerations on Combining Unfolding Inverters with Partial Power Regulators in Battery–Grid Interface Converters

**Ilya A. Galkin [1,\*], Rodions Saltanovs [1], Alexander Bubovich [1], Andrei Blinov [2] and Dimosthenis Peftitsis [3]**

1   Faculty of Electrical and Environmental Engineering, Riga Technical University, LV-1048 Riga, Latvia
2   Department of Electrical Power Engineering and Mechatronics, Tallinn University of Technology, 19086 Tallinn, Estonia
3   Department of Electric Energy, Norwegian University of Science and Technology, 7034 Trondheim, Norway
\*   Correspondence: gia@eef.rtu.lv

**Abstract:** The application of electrochemical cells as a source unit of electrical energy is rapidly growing—used in electric vehicles and other electric mobility devices, as well as in energy supply systems—as energy storage, often together with renewable energy sources. The key element of such systems is the power electronic converter used for DC energy storage and AC grid interfacing. It should be bidirectional to charge and discharge the battery when it is necessary. Two-stage battery interface converters are the most common; their DC-DC stage controls the battery current and adjusts voltage, but the DC-AC stage (inverter or frontend) controls the current in the grid. The use of unfolding inverters in two-stage battery interfaces can have some advantages. In this case, the DC-DC converter produces half-sinewave pulsating voltages and currents, but the unfolding circuit changes the polarity of the voltages and currents and produces no switching losses. Another trend of modern power electronics is the principle of partial power processing. In this case, power electronic converters deal only with a part of the total power; therefore, losses in such converters are reduced. This paper considers combining unfolding frontends with partial power DC-DC converters that enable the further reduction in losses. In this paper, it is shown that such implementation of the partial power conversion principle in semi-DC-AC systems is really possible based on the real-time matching of the voltage of the partial-power DC-DC converter, battery voltage (which depends on its state of charge) and the rectified instantaneous voltage of the AC grid.

**Keywords:** battery energy storage systems; electric vehicles; battery chargers; AC-DC power converters; DC-AC power converters; inverters

## 1. Introduction

Electrochemical cells are historically one of the most known and one of the most widespread devices for the storage of electrical energy [1,2]. This particularly refers to the rechargeable batteries that are the most suitable source of energy for portable electronic equipment, as well as a convenient source of energy for hand tools and household equipment. Nowadays, recent achievements in battery chemistry, in particular those based on Li-Ion technology, accelerate the improvement of the parameters of rechargeable batteries. Overview papers like [3] regularly report a higher specific energy (several hundred Wh/kg) and power (several kW/kg), operation time (several thousand cycles), charge–discharge efficiency (>95%), as well as the more affordable price of Li-Ion batteries. This brings the technology of rechargeable batteries to such application fields like transport (All-Electric Vehicles—EVs or Battery-Powered Electric Vehicles—BEVs) and energy supply (known as battery energy storage systems—BESSs), in particular, to supply systems with renewable energy sources [4]. At the same time, it must be noted that modern rechargeable batteries are not just mechanical combinations/connections of several electrochemical cells. They also often incorporate electronic circuits or battery management systems for cell balancing,

protection and diagnostics [5,6], and, sometimes, thermal management units that stabilize the temperature of these batteries in an intensive charge/discharge process improve their operation parameters even more [7].

The operation of modern batteries occurs in conjunction with dedicated electronic converters controlling the charge and discharge of the batteries. The overviews of BESS usually emphasize two kinds of converters integrated into BESS: isolated and non-isolated [8,9]. The isolated converters are distinguished from the non-isolated ones by the absence/presence of the full power transformer in the converter. The first group typically contains "safe" low-voltage batteries. At the same time, this definition of safety is rather vague. Reference [9] contains a brief analysis of the regulations applicable to BESSs. It has been concluded that these regulations define the constraints for various grid-tied electrical equipment but do not explicitly limit the voltage of batteries. For this reason, in some cases, BESS manufacturers refer to other standards that regard other equipment with batteries, for example, telecom centers [10] and personal mobility vehicles like wheelchairs [11,12]. It is quite typical that these standards separate the parts of the equipment accessible by ordinary users (batteries) from the parts of the equipment accessible only by qualified staff (chargers). While the chargers have quite high AC limitations (for example, 250VAC in [11]), more accessible batteries have much lower DC limitations; ref [10] defines the dangerous level at 60 V while [11] defines it at and 36 V (for lead-acid batteries), and [12] at 50 V (for Li-ion batteries). As a consequence, according to Section 3 of [9], the market-available BESSs typically include a "safe" 48–60 V battery or a battery linked to the ratified grid voltage (300–400 V) or supplied in two configurations with low or high-voltage battery.

The converters of the first type typically include a grid frequency or high-frequency-isolating transformer that galvanically separates the battery component of energy storage from its grid component. For the same safety reasons, the battery interface converters (chargers) of BEVs are also typically isolated. To the same extent, this refers to the converters allocated outside the BEVs—off-board chargers—[13] or those placed inside of them—on-board chargers [14]—as well as the chargers larger [13,14] or smaller [15,16] BEVs.

The converters of the second type, or non-isolated converters, can link to the grid's rather high-voltage batteries. On one hand, the BMS of such batteries is more complex, expensive and less reliable, but on the other hand, this eliminates the need for a full-power-isolating transformer and the corresponding losses. In addition, the same operation power is achieved at lower currents and, therefore, with lower conduction and switching losses. These converters and BESS, therefore, are potentially more energy efficient. Some BESSs available on the market are offered with low and high-voltage battery versions that prove the prospects of this combination of batteries and chargers [17].

When talking about BESS interface converters for high-voltage batteries, it is also necessary to outline their two main topologies: single- and two-stage. The single-stage converters link the floating voltage of the battery and the AC voltage of the grid through a monolithic power converter (grid frontend). They are typically extremely efficient for one operation point with a particular state of charge (SoC) of the battery but not so efficient if SoC is different. Introducing a pre-regulator compensates for the floating of battery voltage and stabilizes the DC voltage of the frontend at the level optimal for its operation. This makes the operation of the frontend more efficient for a wider range of SoC, but it also reduces the efficiency at the best operation point [8,9] due to the extra losses in the pre-regulator.

The final choice of the battery and its interface converter is made based on the trade-off between lower losses and the higher efficiency of BESS on one hand, but on the other, it must consider a safer low-voltage battery, a more simple and reliable battery management system, as well as the simpler installation and maintenance of the BESS.

This work is devoted to a loss reduction in BESSs. Usually, the loss reduction and higher efficiency of BESSs with non-isolated converters refer to two matters: (1) the battery of such a BESS has a higher voltage and, therefore, the interface converter and BESS, on the whole, operate at lower currents leading to lower conduction losses; (2) the absence of the

full-power isolating transformer excludes all the losses associated with this transformer. In addition to these two considerations, this research also utilizes two promising loss reduction techniques. One of them is partial power conversion, but the other one is the use of unfolding inverters. While separate studies on these techniques are frequent, their combination is not studied well. Quite rare reports are devoted to unidirectional coverers, mostly for photovoltaic applications (see Section 2 for details). This paper expands the study to bidirectional systems, namely, to BESSs. Combining these two techniques allows the use of a lower voltage battery without the use of the full-power isolation transformer that causes the above-mentioned trade-off. One part of this research is performed experimentally in order to prove the feasibility of the proposed BESS interface converter. The other part of this work treats a mathematical model of the converter with the goal of determining its losses and the link between its losses and partiality. In both cases, the main research method is data gathering and analysis.

## 2. Approaches to Loss Reduction

The two-stage interface converter for BESS, considered in this paper, is intended for high-voltage batteries, that allow it to operate with lower currents and, therefore, lower conduction losses. In addition, this converter logically combines and utilizes two trends in the design of power electronic converters, also facilitating loss reduction. The first feature is the use of partial power converters in DC systems which, together with a reduction in processed power, also reduces the losses. In turn, the second feature is related to the operation of the network frontend of the two-stage converters. An alternative to the traditional pulse mode operation of the frontend exists is, in this case, the grid–frequency direct commutation of the DC-link to the grid that requires semi-sinusoidal voltage in the DC-link but allows the almost pure elimination of the switching losses (such converters are known as unfolding inverters or unfolders). Let us consider these two trends in more detail.

In contrast to the full-power DC-DC converters that are subject to full input and output voltages, they conduct a full current and, therefore, process the full system power, and partial-power DC-DC converters (PPC) are connected between system inputs and outputs in such a way that the converter processes only the difference between input and output voltages and currents. For this reason, PPCs deal with only a part of the full system's power while its major part is transmitted from the input of the system to its output without any conversion [18,19].

PPC benefits include the following: (1) a lower converter switching current and voltage, which allows the selection of cheaper and more compact semiconductor switches (transistors and diodes); (2) lower converter losses (determined by lower-rated power) which significantly improves the total energy efficiency and facilitates cooling. The lower the voltage/current difference handled by a PPC, the more pronounced the benefits of the partial power conversion. PPCs are particularly convenient to compensate for the parameter floating of DC energy sources or storages, such as batteries or PVs. For example, voltage reduction in a battery together with its SoC at its discharge may be compensated by a PPC, the input of which is connected to the battery in parallel with its output in a series, thus forming the sum of the battery and PPC voltages. PPC then generates the difference between the maximum battery voltage (at SoC 100%) and its actual voltage. So, by definition, such PPC processes the mentioned difference that is lower than the full battery voltage. For a 100-cell Li-Ion battery, the maximal PPC voltage can conclude $+100 \times (4.2 - 2.5) = +170$ V (where 2.5 V is the cut-off voltage of the cell, but 4.2 V is the maximal open circuit voltage of the fully charged cell) while the full-scale converter deals with 420 V. In this example, the full power is 2.5 times higher than the partial processed power.

Unipolar PPCs can only add (or subtract) their voltage (current) to the base value and bipolar PPCs are capable of both adding and subtracting. The PPCs of the second kind can operate with twice as low voltage/current/power compared to single-polarity PPCs [20]. For example, in the previous battery example, bipolar PPC can operate with

$\pm 100 \times (4.2 - 2.5)/2 = \pm 85$ V while providing, at the same time, the complete compensation of the battery voltage drop at 170 V. In this case, the processed power is five times lower than the actual power.

Schematically, PPC can be a DC-DC converter of any isolated topology. The most versatile implementation of unipolar PPCs is typically based on a Double Active Bridge (DAB), but other schemes are also possible. For example, ref. [21] presents PPCs of flyback and full-bridge phase shift topologies.

Bipolar PPCs include bidirectional or four-quadrant switches at their secondary side. For example, ref. [22] presents a bipolar version of the full-bridge phase shift converter from [21], while [23] describes a PPC with a bipolar DAB. Other schemes utilize resonant chains for better commutation and lower losses; for example, a bipolar DAB with a resonant tank is reported in [24]. It must be noted that all the above-mentioned PPCs [20–24] are intended solely for use in DC-DC systems.

The second trend in the field of power electronic converters, which is utilized in the considered system, refers to the principles of synthesis of the AC voltage in two-stage DC/AC inverters. As has been mentioned, the first stage of such inverters is a DC/DC regulator, while the second stage is a network frontend (rectifier/inverter). These stages are connected through a DC-bus. Traditionally, both stages are pulse mode converters—the first compensates for changes in the battery, and the second forms a sine-form voltage and connects it to the grid. The alternative method of synthesis of the AC voltage/current assumes that the regulator not only compensates for the voltage changes in the battery but also forms a semi-(rectified) sinewave at its output. As a result, the DC-link voltage and current pulsate, while the frontend just unfolds these pulses to the grid with predetermined polarity and with a low network frequency, operating as a commutator or as diodes in a diode rectifier [25,26]. It is clear that the power losses in such a commutator (known as unfolding frontend or unfolder) are lower because they do not include the component of the losses associated with high-frequency switching. This principle of commutation of the pre-shaped semi-sinewave voltage is also applicable to three-phase systems [27,28].

When considering the two-stage battery interface inverters with a pulsating DC-bus, one can notice that, while the frontend produces lower losses, the regulator forms the voltage in the full range from zero to the amplitude of the network voltage that is hardly compatible with partial power principle. On the other hand, if the DC-bus is stabilized, then the regulator can process partial power and may have lower losses, but the frontend is a pulse mode circuit with additional switching losses. It is quite logical that certain attempts were made in order to combine the PPC principle with a pulsating DC-bus and unfolding frontend.

One of the earliest distinct attempts to achieve power partiality with unfolders is reported in [29,30]. The papers present a two-stage two-level voltage-sourced inverter for PVs that adds the variable voltage from the pulse mode circuit to the constant voltage of PV. These works aim to compensate for the voltage reduction over the PV matrix that leads to the operation of the inverter at limited power if the PV voltage is low. Being a unidirectional inverter, it is suitable only for PVs. In addition, the presented inverter is not a truly partial power converter but just operates at reduced parameters (a good explanation of this phenomenon can be found in [19]).

Paper [31] presents a two-stage interface converter for batteries with a mid-point, which combines a specific unipolar two-level inverter with an explicit unfolder. In this case, the converter transforms the constant voltage from one or another DC source (fractions of the battery) into a semi-sine voltage of the DC-bus that is applied to the grid by the unfolder. The inverter rater utilizes the principle of fractional power conversion when partial power is taken from a distinct fraction of the power source (that could lead to the non-even aging of the battery cells). The paper itself is more focused on control matters, particularly, on the problem of voltage zero crossing, but its loss analysis is very brief.

Paper [32] and patent [33] present a two-stage inverter, which combines the true PPC principle with a pulsating DC-bus and unfolding frontend. This work utilizes a series

input–parallel output (SIPO) PPC scheme and, like [29,30], is intended for PV interfacing. These documents provide a feasible study of SIPO PPC with the UF inverter and DC allocation of the current firming inductor. At the same time, they do not pay much attention to the study of the actual partiality and its influence on the losses and parameters of the semiconductor switches. In addition, the proposed technical solutions are not suitable for use with BESS due to their unidirectional nature.

The essence of this work follows from the mismatch of the above-discussed solutions [29–33] from considering the application or principle of the true partiality of power conversion. Its main contribution includes the development of a novel BESS power interface and corresponding control, as well as their evaluation from the point of view of energy efficiency and parameters of the switches. The first novelty, therefore, is a new power electronic converter for BESS that combines a bidirectional unfolder, bidirectional parallel input–series output (PISO) partial power converter and pulsating DC-link. An essential part of this novelty is the method of interfacing, which processes the floating voltage of the BESS battery, forms pulsating semi-sine voltage in the DC-link, and applies it to the AC-grid. The second novelty of this work is a simplified quick methodology of the loss evaluation of the proposed converter based on its actual real-time partiality ratio depending on operation conditions (SoC and grid phase). At last, this work briefly evaluates the positiveness of the proposed BESS interface from the point of view of the voltage and current stress on its switches.

## 3. Outlines of Proposed Two-Stage BESS Interface Converter

### 3.1. Structure of Converter

The most explicit configuration of BESS with the proposed two-stage converter is presented in Figure 1a. Apart from the battery and grid, it contains an unfolding grid frontend (UF—inverter, operating at grid's frequency), an isolated bidirectional PPC capable of generating bipolar voltage (PPC$_{chg,dis}$), as well as a "virtual" DC-bus (qDC) with semi-sinewave pulsating voltage. The voltage of the battery is approximately twice as low as the amplitude of the grid voltage. Due to this, the PPC is connected with the battery in series on the DC-bus side and in parallel on the battery side. This configuration (named in [32] as PPC Type II) is analyzed in the present work, in contrast to PPC Type I, which is mostly studied in [32]. Further, this series connection of the battery and PPC is attached in parallel to the UF inverter. This may be an ordinary single-phase H-bridge as in [25] or a three-phase circuit like in [28]. In turn, the PPC can be constructed as any isolated bipolar bidirectional circuit, including the circuits with resonant tanks; for example, a bidirectional DAB is presented in [34].

Another configuration of BESS with the PPC and UF inverter is shown in Figure 1b. It includes an isolated bidirectional unipolar PPC, a "pulsating" DC-bus (qDC+), and an additional unfolding inverter (UF+). The unipolar PPC with the additional inverter UF+ operates as the bipolar PPC of the previous configuration. This reduces number of the switches operating in the high-frequency mode and, therefore, the corresponding switching losses.

One more improvement in the initial BESS configuration assumes the splitting of the bidirectional PPC into two unidirectional PPCs. One of them operates only in the battery charging mode, while another one operates during battery discharge. This means the absence of bidirectional switches that reduce the number of semiconductor elements in each current loop and the corresponding conduction losses. This BESS's configuration is shown in Figure 1c.

Finally, the combining of the two above-mentioned improvements, i.e., the splitting of the bidirectional PPC into two unidirectional ones and the use of unipolar PPC with the extra unfolder instead of the bipolar PPC, provides the achievement of their benefits together. In addition, such configuration enables the fine-tuning of the design of these separated parts of the two-stage BESS interface converter.

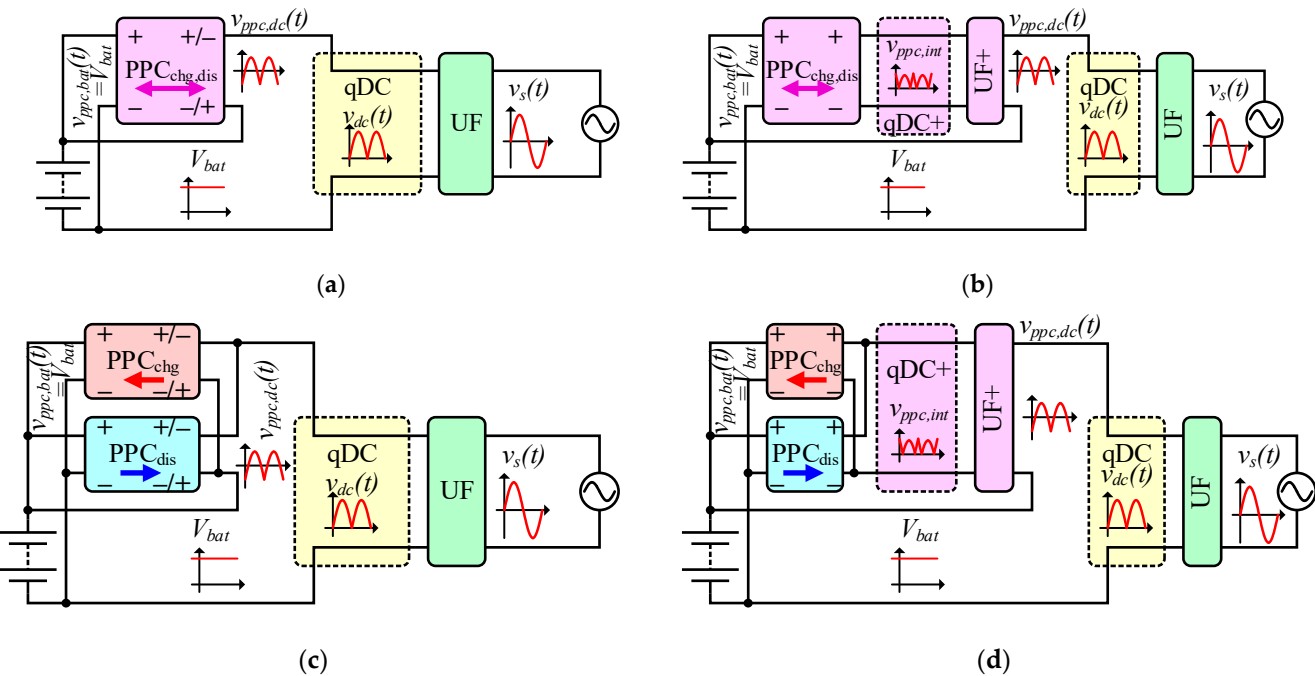

**Figure 1.** Structural diagrams of a two-stage converter with unfolding frontend and partial power DC/DC converter: (**a**) basic configuration, (**b**) configuration with additional unfolder, (**c**) configuration with two converters dedicated for battery charging and loading and (**d**) configuration with additional unfolder and two converters.

### 3.2. Topologies and Operation of Frontend

The frontend (Figure 2) is composed of a commutation matrix, switching at grid frequency, and an inductance coil that serves as a current-forming element and can be allocated at the DC or AC port of the frontend.

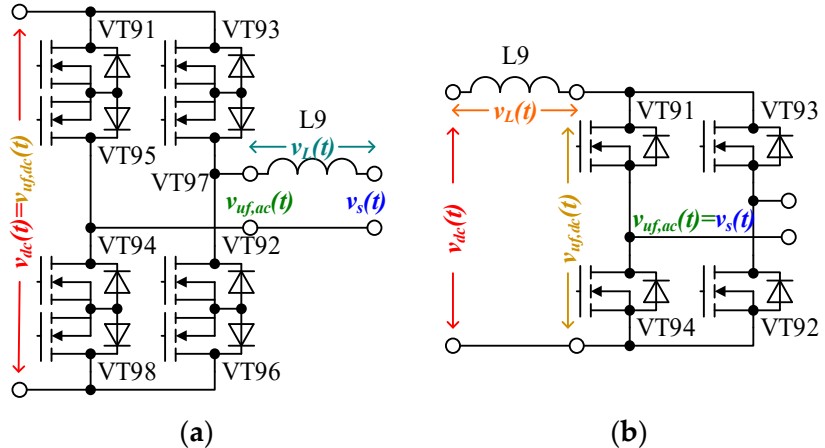

**Figure 2.** Generalized schematics of unfolding inverters (single phase, basic elements): (**a**) with inductor at AC port and (**b**) with inductor at DC port.

When the coil is allocated at the AC port (Figure 2a), its first contact is fixed at the grid, while the second one is connected through the commutation matrix to the DC-link. The coil then operates with alternating sine current and voltage. Since the voltage in the grid (Figure 3a—blue) must be in line with the grid current (Figure 3b) formed in the coil, the voltage of the coil (Figure 3a—magenta) must have a ±90° shift. Therefore, the voltage at the first end of the coil (at the AC port of the commutation matrix) must be slightly leading (for the battery loading mode, as in Figure 3a—green, or lagging, for battery charging mode)

and slightly higher, compared with the grid voltage on the second end of the coil. This, in turn, means that the commutation matrix must be a four-quadrant converter, capable of conducting a current in both directions at both polarities of the voltage. Due to the doubled number of transistors and more complicated control, this case is out of practical interest except for autonomous loads like the motors of larger or smaller vehicles. In a similar way, Figure 3c,d represents the battery charging mode, when the voltage at the AC port of the commutation matrix is lagging, but the current in the coil and grid has a 180° shift (is negative) compared to the grid voltage.

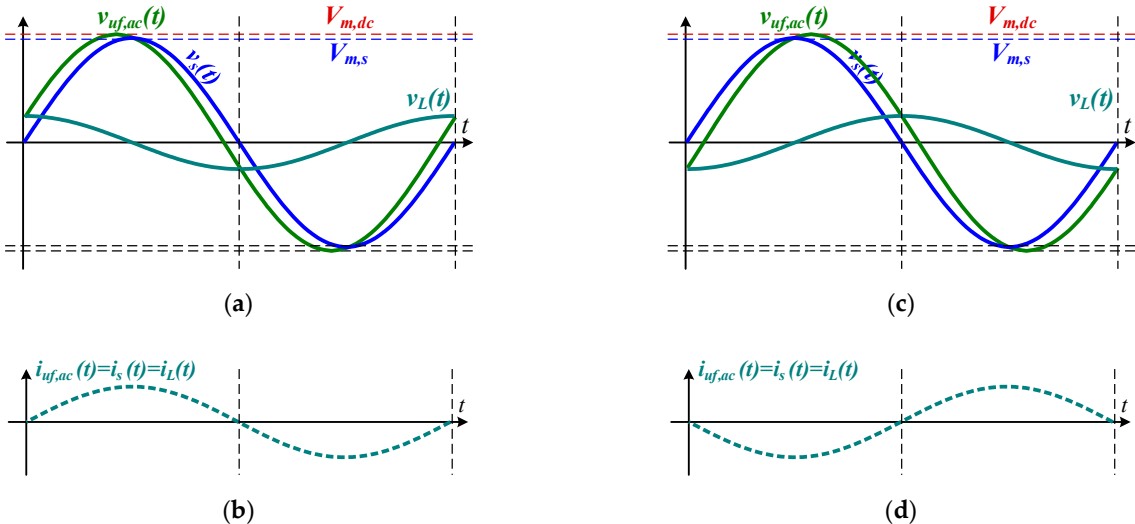

**Figure 3.** Operation of inductance in unfolding inverter with inductor at AC port: (**a**) voltages and (**b**) current (battery loading mode); (**c**) voltages and (**d**) current (battery charging mode).

The alternative allocation of the current limiting and forming inductance coil is at the DC port of the commutation matrix. Then, the first end of the coil is attached to the DC-link voltage ($v_{dc}(t)$ in Figure 4a), but at the second, the matrix forms a semi-sinusoidal grid voltage ($v_{dc,uf}(t)$ in Figure 4a). Then, the voltage over the coil is semi-sinusoidal with 90° (Figure 4a—orange) as well as its current (Figure 4b). With such a configuration, the polarity of the voltage at the AC port of the frontend always corresponds (must) to the polarity of the current. Therefore, the commutation matrix may be a common transistor H-bridge (Figure 2b). Similarly, Figure 4c,d represents the battery charging mode, when, within any halfwave, the voltage at the DC port of the grid commutation matrix is lagging, but the current in the coil and DC-link is negative.

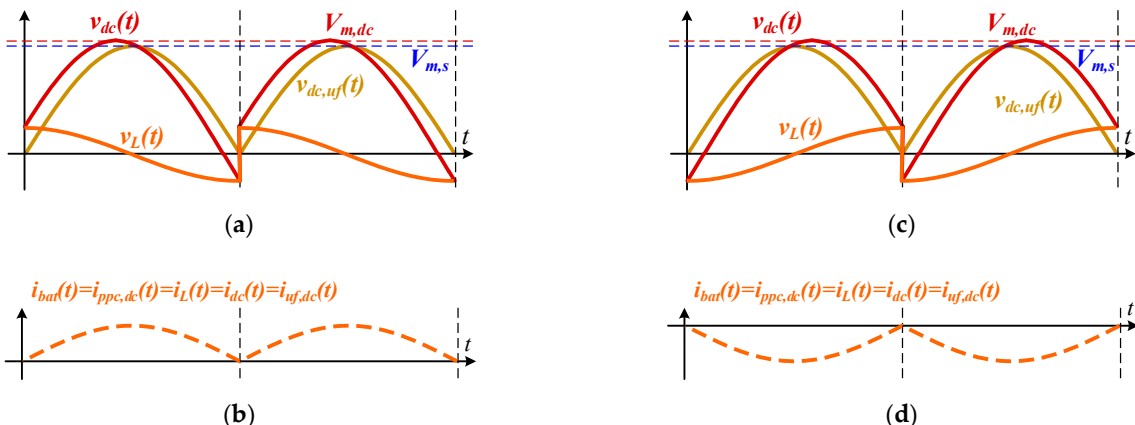

**Figure 4.** Operation of inductance in unfolding inverter with inductor at DC port: (**a**) voltages and (**b**) current (battery loading mode); (**c**) voltages and (**d**) current (battery charging mode).

In both cases, the DC-link must provide semi-sinewave voltage, composed of sine fragments with a certain small angle from γ to 180 + γ. (for the battery loading mode) and from 180 − γ to −γ (for the battery charging mode).

### 3.3. PPC Topology and Operation

The second part of the considered battery interface system is a DC-DC converter in the partial power processing scheme with one port connected to the battery in parallel and another in series. As has been mentioned above, it may be based on any isolated converter capable of generating bipolar voltage at the port connected to the battery in series. Two options have been considered within this work. The first one is a step-up/down PPC with bipolar DAB (BDAB) that includes a 2 × 2 matrix of bipolar switches described, for example, that presented in [23] and Figure 5a. Another one is a step-up/down PPC with a standard DAB followed by an extra transistor bridge, serving as a polarity toggler (one more unfolder), as shown in Figure 5b. Due to the twice lower switching losses in the bipolar part, the latter-mentioned configuration is taken as the base for further experimenting and analysis.

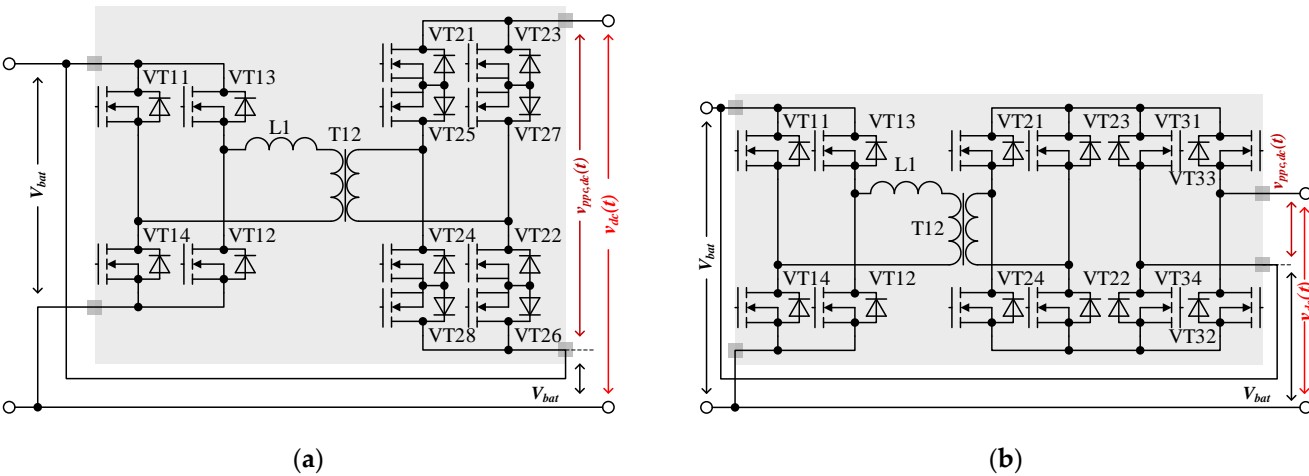

**(a)**　　　　　　　　　　　　　**(b)**

**Figure 5.** Considered configurations of partial power converters: (**a**) with bipolar DAB; (**b**) with bipolar DAB and extra unfolder.

As follows from the previous section, the main function of the DC-DC stage is forming a semi-sine voltage in the DC-link of the converter.

For both configurations of the frontend, this voltage in the battery loading mode is composed of the semi-sine half-waves that are slightly leading compared with the rectified grid voltage, i.e., they start from some small phase γ and continue to the angle 180 + γ. Then, the resulting current is passed to the grid, but the battery of the BESS is loaded.

In contrast, in the battery charging mode, these semi-sinusoidal half-waves of voltage must be lagging compared to the rectified grid half-waves, i.e., they start from 180 − γ and continue to −γ. Also, in this case, the operation of the DC-DC converter does not depend on the kind of the frontend.

The accurate forming of the current requires a slightly higher amplitude of these half-waves formed by the DC-DC converter. This amplitude can be found from the right triangle of the voltages, the legs of which are the grid voltage and the coil voltage, but the hypotenuse is the voltage of the DC-link. Then,

$$V_{m,dc} = \sqrt{V_{m,s}^2 + (2\pi f \cdot L_s \cdot I_{m,s})^2} \tag{1}$$

$$\text{and } \gamma = arctg\left(\frac{2\pi f \cdot L_s \cdot I_s}{V_{m,s}}\right) \tag{2}$$

Here, $V_{m,s}$—the amplitude of the grid voltage, $I_{m,s}$—the requested amplitude of the grid current, and $L_s$—the inductance of the coil.

The expected operational diagrams of the converter are given in Figure 6 (the BESS discharge or loading mode), Figure 7 (the BESS charge mode) and Figure 8 (the discharge mode to an autonomous load). The "pulsating" DC-link of the interface converter links the unfolding grid frontend VT9x with the series-connected battery and DC port of the PPC. This is why the semi-sinusoidal half-waves (brown curve in these figures), passed to the grid through the unfolding inverter VT9x, are formed as a sum of the battery voltage (black curve) and PPC voltage (red curve). On the other hand, the voltage of the PPC can be found as the difference between the desirable DC-link voltage and battery voltage, i.e., this voltage contains the same semi-sine half-waves with the negative offset, which is equal to the battery voltage. If the PPC operated as a buck converter, the discharged battery (SOC close to 0%) must provide 50% of the voltage span in the DC-link which, in a general case, can be found as follows:

$$V_{span,dc} = V_{\max,dc} - V_{\min,dc} = V_{m,s} + V_{m,s}\sin(\gamma) \tag{3}$$

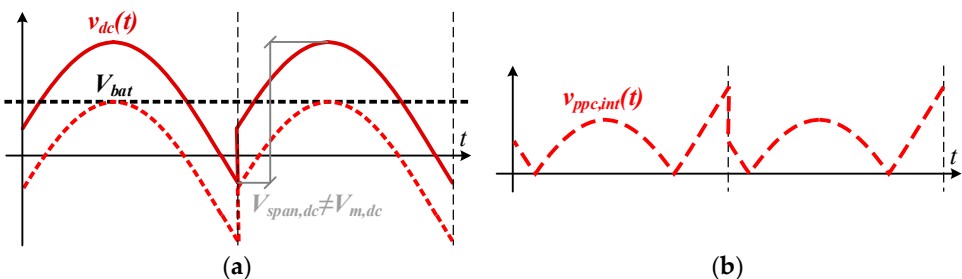

**Figure 6.** Voltage diagrams in BESS discharge mode: (**a**) voltages, typical for DC-link (black—battery voltage, red—PPC voltage at DC-link, brown—DC-link voltage); (**b**) internal voltage of PPC before extra unfolder.

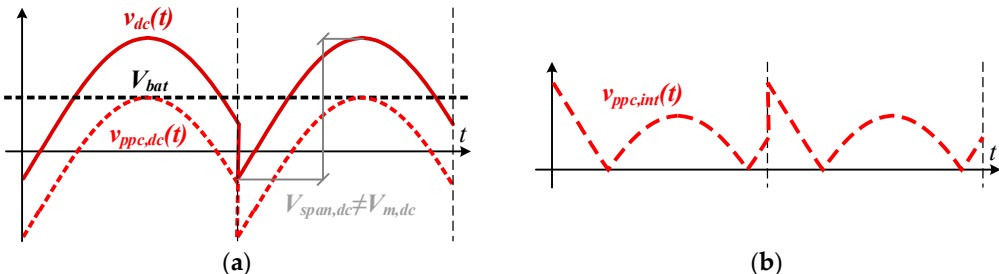

**Figure 7.** Voltage diagrams in BESS charge mode: (**a**) voltages typical for DC-link (black—battery voltage, red—PPC voltage at DC-link, brown—DC-link voltage); (**b**) internal voltage of PPC before extra unfolder.

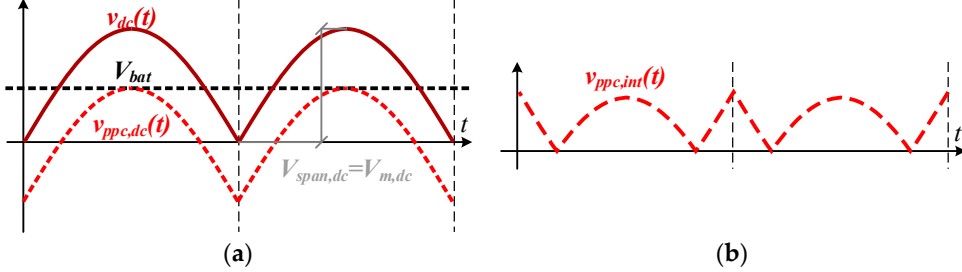

**Figure 8.** Voltage diagrams in BESS discharge mode for an autonomous ohmic load: (**a**) voltages, typical for DC-link (black—battery voltage, red—PPC voltage at DC-link, brown—DC-link voltage); (**b**) internal voltage of PPC before extra unfolder.

If the PPC is built as a DAB with an extra unfolder (Figure 5b), then, (1) firstly, the DAB generates the rectified form of this voltage (Figures 6b–8b) and then (2) the extra unfolder VT3x applies it to the DC port of the PPC with the required polarity. Meanwhile, the current through the PPC and the battery remains semi-sinusoidal, as shown in Figure 4b.

## 4. Experimental Validation of Converter

In order to verify the proposed concept, an experimental setup with a rated power of 250 W watts was assembled. Its schematic is given in Figure 9 and the experimental prototype is depicted in Figure 10. The main components of the prototype are listed in Table 1.

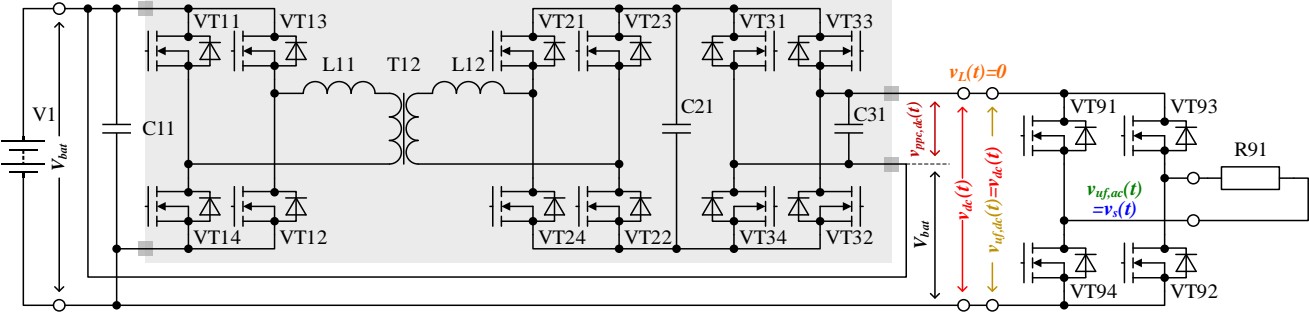

**Figure 9.** Schematic of the experimental setup with unfolding grid frontend, bipolar DAB and extra unfolder.

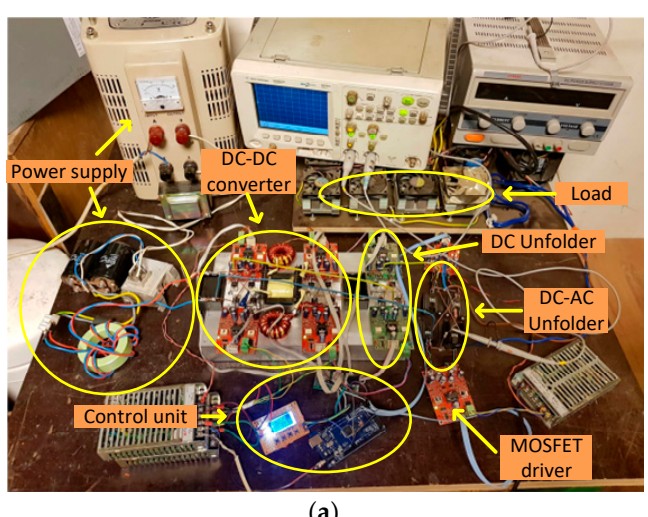

(**a**)

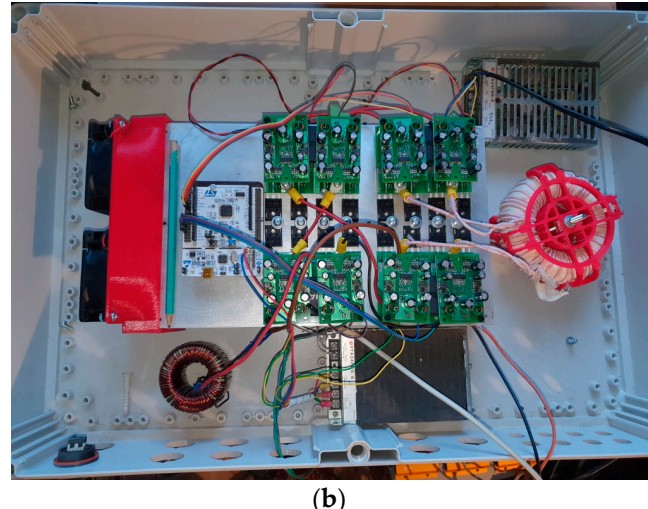

(**b**)

**Figure 10.** Layout of the initial (**a**) and final (**b**) experimental setups.

The main switching elements (IPP60R040C7) are mounted on an aluminum heatsink with natural convection. The isolation transformer T12 of the partial power DC-DC converter (PPC) utilizes the turn ratio of the primary and secondary windings as 1:1. The primary and secondary windings of the transformer contain 25 turns of the CLI 120 × 0.1 face wire. Two E-shaped ferrite profiles ETD59 made of 3C94 with permeability μ = 2300 were used as the transformer core. Chokes L1 and L2 (the split inductor of the DAB) were wound on powder iron rings with the same CLI 120 × 0.1 face wire and had an inductance of 60 μH. At the input, the output and in the middle of the PPC 30 μF film capacitors are located. MOSFET drivers based on the ACPL-333J microcircuit provide the opening and closing of power transistors using a digital signal from the microcontroller through an optocoupler. The drivers have current protection. The control unit of the converter is based on ATmega2560 MCU, which has been selected due to its 12-channel, 16-bit PWM module. A power supply was applied as a battery for the quick imitation of different SOCs. The

load of the converter is an autonomous AC load, represented by a 210 Ω resistor that results in 250 W of the output power generated at 230 VAC. The setup refers to the operation principles given in 0.

**Table 1.** Components of the experimental setup.

| Symbol | Component | Manufacturer, City and Country | Remark |
|---|---|---|---|
| VTxx | IPP60R040C7 | Infineon Technologies AG, Neubiberg, Germany | nMOSFET, Si, 650 V, 40 mΩ |
| T12 | Custom | | 1:1, windings—25 turns of CLI 120 × 0.1 face wire, 2 ETD59-3C94 with μ = 2300 |
| L11, L12 | Custom | | 60 μH, core—powder iron rings, 17 turns of CLI 120 × 0.1 face wire |
| C11, C21, C31 | MKP1848S | Vishay Intertechnology Inc., Malvern, USA | Metallized Polypropylene Film Capacitor, 30 μF, 1000 V |
| - | ACPL-333J | Broadcom Inc., San Jose, USA | 2.5 Amp Output Current IGBT Gate Driver with Integrated Desaturation Detection, Miller Clamp and Fault Status Feedback |
| - | ATmega2560 | Microchip Technology Inc., Chandler, USA | MCU with 12-channel of 16-bit PWM module. |
| R91 | 4 × SLN175J230E, 230 Ω, 1 A | Ohmite, Warrenville, USA | High Power Resistor |

The measurements obtained from the setup are presented in Figure 11. When considering these diagrams, it must be taken into account that the voltage sensors utilize 1:50 resistive dividers. In turn, all currents were taken from a 1 Ω resistor providing a current scale of 1 V per 1 A. The results given generally correspond to the expected operational diagrams given in Figure 11. Therefore, the general idea of the operation of the proposed converter was confirmed.

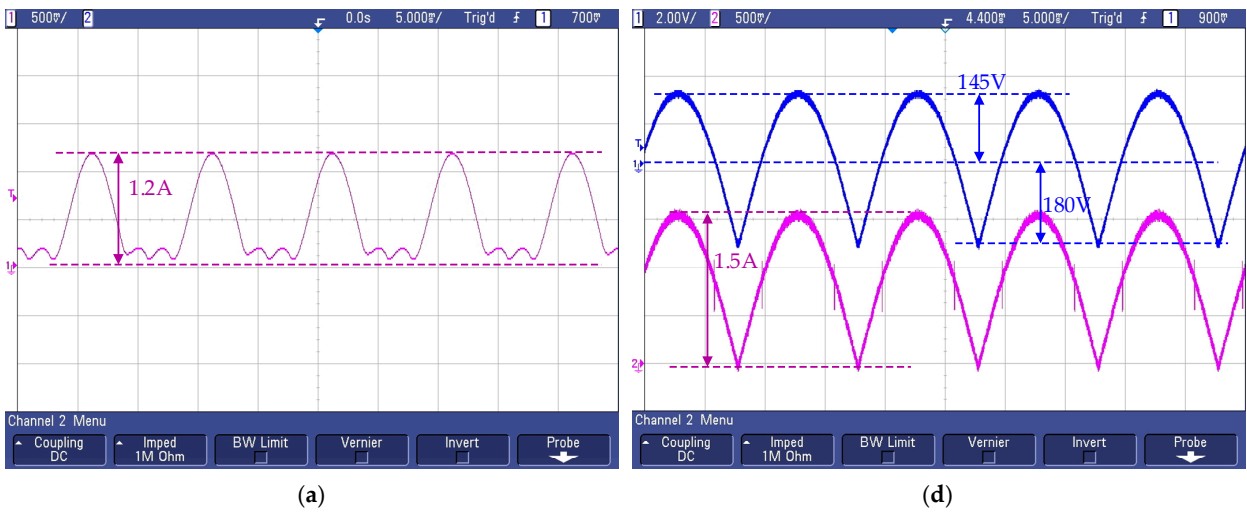

(**a**)      (**d**)

**Figure 11.** *Cont.*

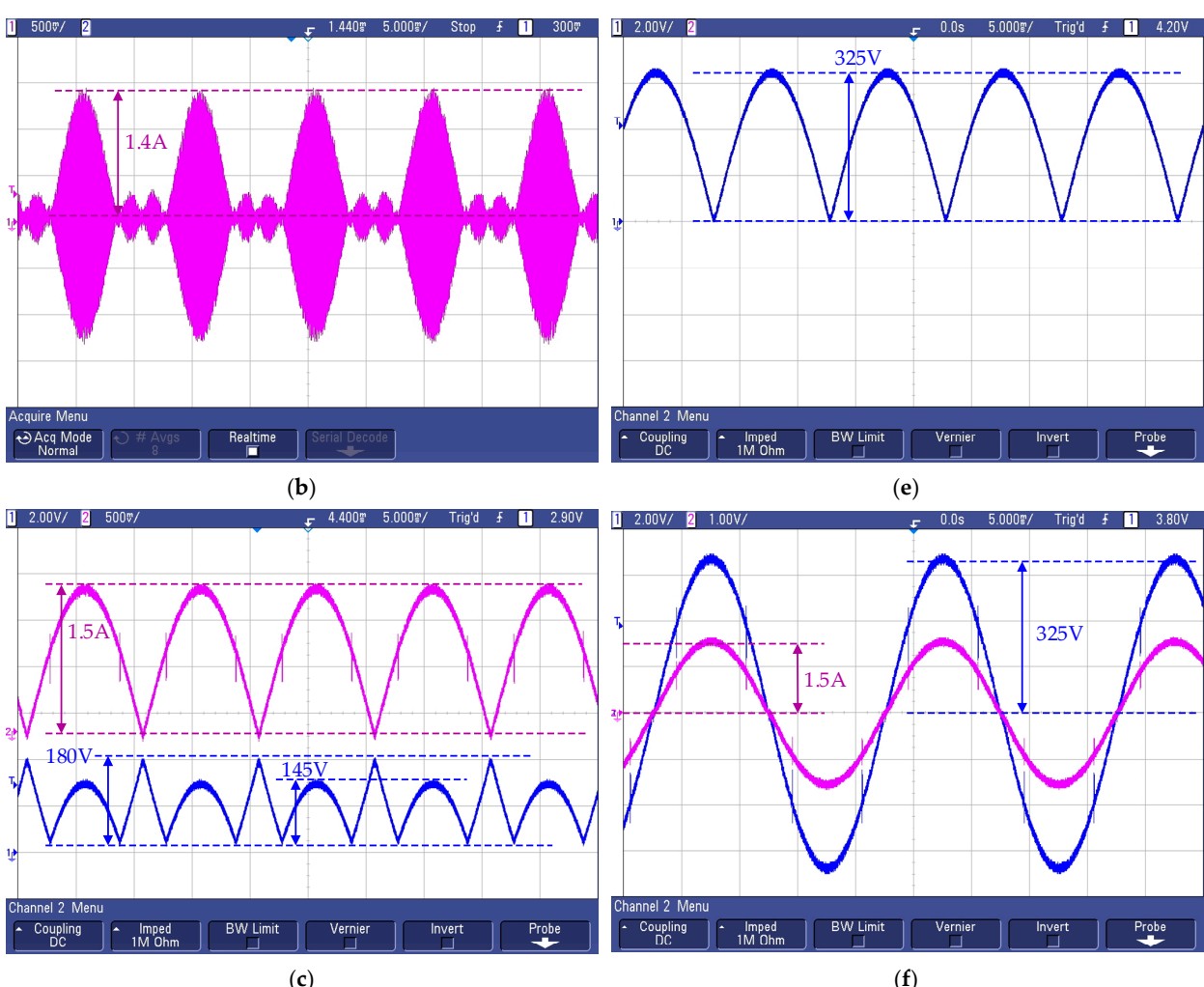

**Figure 11.** Test results of the proposed BESS interface: (**a**) PPC current at the battery side $i_{ppc,bat}(t)$; (**b**) PPC current in the transformer $i_{ppc,tr}(t)$; (**c**) PPC internal voltage $v_{ppc,int}(t)$ and current $i_{ppc,int}(t)$ before the extra unfolder; (**d**) PPC voltage $v_{ppc,dc}(t)$ and current $i_{ppc,dc}(t)$ at the side of the DC-link; (**e**) DC-link voltage $v_{dc}(t)$; and (**f**) voltage $v_s(t)$ and current $i_s(t)$ in the grid (load).

## 5. Considerations on Partiality and Its Actual Influence

### 5.1. Evaluation of Partiality

When evaluating the partial power converters, it is important to determine which part of the total power is actually processed by the converter. The lower the part is, the higher the potential energy efficiency. This part sometimes is expressed in % relative to the full power—as a power partiality ratio:

$$PPR = \frac{P_{PPC}}{P_{\Sigma}} \cdot 100\% \tag{4}$$

Here, $P_{PPC}$ is the active power of the PPC and $P_{\Sigma}$ is the total power of the system. In contrast to the DC systems, these parameters of the proposed BESS interface converter are integral values. The full system's power can be calculated at the grid side of the converter. With proper control (i.e., without harmonic distortions and reactive power), the full power is found as a product of the RMS grid voltage $V_s$ and grid current $I_s$. In turn, the active power of the DC-DC converter can be found as a form of average instantaneous power at any port of the converter—either $p(t)_{ppc,bat}$, $p(t)_{ppc,int}$ or $p(t)_{ppc,dc}$. The power at the DC port ($p(t)_{ppc,dc}$) is easy to express because, in the case of the proper control, the current $i_{ppc,dc}(t)$ is a rectified sine-wave (Figures 4b and 11f). In turn, $v_{ppc,dc}(t)$ is the difference of the DC-link

voltage $v_{dc}(t)$ and battery voltage $V_{bat}$. Considering the above-mentioned and applying the network voltage $v_s(t) = V_{s,\max} \cdot \sin(\omega t)$ with the network current $i_s(t) = I_{s,\max} \cdot \sin(\omega t)$ the instantaneous power is defined as

$$p_{ppc}(t) = I_{s,\max} \sin(\omega t) \cdot (V_{s,\max} \sin(\omega t) - V_{bat}) \tag{5}$$

but its average value (active power through the DC-DC converter) is described as

$$P_{ppc} = \frac{1}{T} \int_0^T I_{s,\max} \sin(\omega t) \cdot (V_{s,\max} \sin(\omega t) - V_{bat})\, dt \tag{6}$$

One part of this integral (minuend) is the network power $P_s = I_s \cdot V_s$, but another part (subtrahend) can be found as the integral of a half-sine.

$$\frac{1}{\pi} \int_0^\pi I_{s,m} \sin(\theta) \cdot V_{bat} d\theta = \frac{2}{\pi} \sqrt{2} I_s V_{bat} \tag{7}$$

Then, the PPR is reversely proportional to the battery voltage:

$$PPR = \left(1 - \frac{2\sqrt{2} V_{BAT}}{\pi V_{s,rms}}\right) \cdot 100\% \tag{8}$$

Let us calculate the PPR for the step-down DC-DC converter and battery providing at least half of the grid voltage amplitude (about 160 VDC) when it is discharged (64 Li-Ion cells with voltage 160–230 VDC). These numbers are given in Table 2. It is seen that (8) expresses a linear function that shows the diminishing influence of the battery voltage on PPR. At $V_{bat}$ = 255 V, which is possible during battery charging for a fully charged battery, (8) shows no PPC contribution in power transfer. This corresponds to an explicit reduction in PPC power even down to 0 in DC systems at no voltage (or current) difference between the ports of the converter. Moreover, the further increase in the battery voltage and forming of the semi-sine half-waves with the help of the DC-DC converter leads to negative power through the DC-DC converter. Therefore, the above-mentioned value represents the reasonable voltage of the battery.

**Table 2.** Partiality evaluation.

| SOC [%] | $V_{bat}$ [V] | PPR [%] | SOC [%] | $V_{bat}$ [V] | PPR [%] | SOC [%] | $V_{bat}$ [V] | PPR [%] | SOC [%] | $V_{bat}$ [V] | PPR [%] | SOC [%] | $V_{bat}$ [V] | PPR [%] |
|---|---|---|---|---|---|---|---|---|---|---|---|---|---|---|
| 0 | 160 | 37 | 30 | 181 | 29 | 60 | 202 | 21 | 90 | 223 | 13 | >100 | 244 | 4 |
| 10 | 167 | 35 | 40 | 188 | 26 | 70 | 209 | 18 | 100 | 230 | 10 | >100 | 252 | 2 |
| 20 | 174 | 32 | 50 | 195 | 24 | 80 | 216 | 15 | >100 | 237 | 7 | >100 | 256 | 0 |

At the same time, the data presented in Table 2 are only the surficial presentation of the actual power transfer processes in the DC-DC converter; like AC active power, reactive and harmonic power is only a representation of the real-time power consumption in the AC grid. Figure 12 shows the instantaneous power of the converter, calculated according to (5) for 1 kW of the system power and different battery voltages. The declinations of the instantaneous power form its averaged value, used in (5)–(7), which are always significant (even with no actual power in the converter). Therefore, the averaged PPR, expressed by (8), is of limited usability and the actual influence of partiality has to be evaluated over a time span for different parameters of the system.

### 5.2. Influence of Partiality

The influence of partiality is studied in this section based on a mathematical model for a two-stage BESS interface converter with a pulse-mode or unfolding frontend, suitable battery (190–315 V for known configurations and 160–270 V for the proposed combination of UF and PPC) and flyback pre-regulator. The flyback is the simplest isolating DC/DC converter; it is capable of converting voltage in both directions (step-up or step-down) and is more convenient for comparison due to simpler associated calculations.

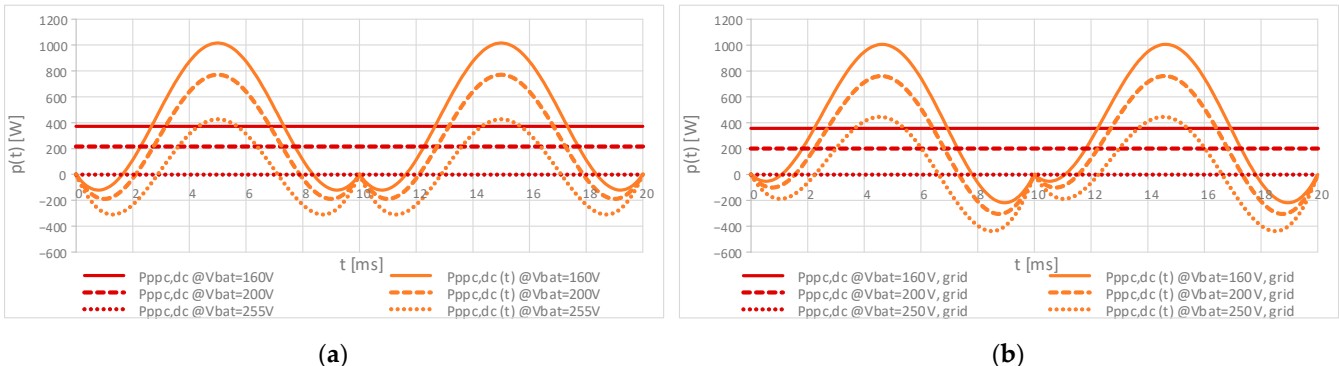

(**a**)                                                              (**b**)

**Figure 12.** Instantaneous power at the DC port of a partial power DC-DC converter (for 1 kW of the total power): (**a**) battery discharge to autonomous load (as considered in Chapter IV); (**b**) battery discharge to the grid.

#### 5.2.1. Brief Loss Evaluation

Typically, the partial power converters are considered a good alternative to the full power systems due to lower voltage and current stress on the switches, as well as due to the lower processing power and, therefore, losses. As shown in Figures 6–8 and 11, the partial power DC-DC converter of the above-proposed BESS interface conducts semi-sine halfwaves of the current with the grid amplitude. As for the commutated voltage of the PPC, it is defined by the battery that, in the following examples, it has a voltage in the range of 160–230 V assuming a step-down DC-DC converter. These values are not much lower than the amplitude of the grid voltage (325 V). Therefore, this typical advantage of the PPC is not so explicit in the case of the considered BESS interface.

In order to evaluate the impact of the proposed configuration on the power losses let us consider a couple of the simplified numerical examples. Firstly, a common BESS interface with a full power pulse-mode frontend and regulator is analyzed.

For the simplified analysis of the switching losses, let us assume the following: (1) the commutated voltage is the voltage of the DC-link that is equal to the amplitude of the grid voltage $V_{s,m} = 325$ V; (2) voltage rise and fall times tv are equal and proportional to the commutated voltage (i.e., they are constant); (3) current rise and fall times $t_i$ are equal and proportional to the commutated current; (4) voltage/current change times are equal in the middle of the semi-sine current half-wave at maximal current; (5) the switching frequency is chosen so that in the middle of the semi-sine half-wave the length of 1 turn-on or 1 turn-off commutation is 1% of the switching period (Relative Duration of Switching RDS); (6) and, for simplicity, the analysis is provided for the battery discharge mode only.

#### 5.2.2. Losses of Full Power BESS Interface

The most common H-bridge topology operates as a grid-to-battery boost converter with the current path provided by a constantly conducting diode and toggling couple transistor–diode. It is, therefore, possible to conclude that in the continuous current mode,

two switches are always conducting a semi-sine current. Then, assuming for simplicity that both of them are p-n devices, their losses can be defined as

$$\Delta P_{FE1,c} = 2 \cdot \frac{1}{\pi} \int_0^\pi \Delta p_c(t) dt = 2 \cdot \frac{1}{\pi} \int_0^\pi V_0 I_{s,m} \sin(\theta) d\theta = \frac{4 V_0 I_{s,m}}{\pi} \tag{9}$$

where $V_0$ is the rated voltage drop over the switch (assumed as a constant for simplicity reasons and equal to 2 V).

The switching losses can be calculated assuming the triangular shape of instantaneous power on the switches during the commutation. Then, the maximal energy loss of turn-on and turn-off commutations is achieved at grid phase 90° when its voltage and current are maximal.

$$E_{FE1,sw,m} = 2 \cdot \frac{V_{s,m} I_{s,m}(t_{i,m} + t_{v,m})}{2} = V_{s,m} I_{s,m}(t_{i,m} + t_v) \tag{10}$$

Then, the maximal equivalent power losses are as follows:

$$\Delta P_{FE1,sw,m} = \frac{V_{s,m} I_{s,m}(t_{i,m} + t_v)}{T_{sw}} = V_{s,m} I_{s,m} RDS = A \tag{11}$$

Since the current change times are considered proportional to the current that is semi-sinusoidal for the other points, the equivalent switching losses are as follows:

$$\Delta P_{FE1,sw,k} = \frac{1}{T_{sw}} \cdot 2 \cdot \frac{V_{sw,k} \cdot I_{sw,k} \cdot (t_{i,k} + t_{v,k})}{2} =$$

$$= \frac{V_{s,m} \cdot I_{s,m} \sin \theta_k \cdot (t_{i,m} \frac{I_{s,k}}{I_{s,m}} + t_v \frac{V_{s,k}}{V_{s,m}})}{T_{sw}} = \tag{12}$$

$$= \frac{V_{s,m} \cdot I_{s,m} \sin \theta_k \cdot (t_{i,m} \sin \theta_k + t_v)}{T_{sw}} = V_{s,m} I_{s,m} \frac{RDS}{2} \sin \theta_k + V_{s,m} I_{s,m} \frac{RDS}{2} \sin_2 \theta_k =$$

$$= \frac{A}{2} \sin \theta_k + \frac{A}{2} \sin_2 \theta_k$$

The total switching losses of the frontend are

$$\Delta P_{FE1,sw} = \frac{1}{0.5 T_s} \sum_{k=1}^N \Delta P_{sw,k} T_{sw} = \frac{2}{T_s} \sum_{k=1}^N \Delta P_{sw,k} \Delta t_k \tag{13}$$

that can then be converted to the integral form

$$\Delta P_{FE1,sw} = \frac{1}{\pi} \int_0^\pi \Delta P_{sw}(\theta) d\theta \tag{14}$$

Applying (12) gives the follows:

$$\Delta P_{FE1,sw} = \frac{1}{\pi} \int_0^\pi \left( \frac{A}{2} \sin \theta_k + \frac{A}{2} \sin_2 \theta_k \right) d\theta = = \frac{A}{\pi} + \frac{A}{4} - 0 = V_{s,m} I_{s,m} RDS \frac{\pi + 4}{4\pi} \tag{15}$$

The above equations for 1 kW of the grid power and with given assumptions produce losses $\Delta P_{FE1,c}$ = 7.8 W and $\Delta P_{FE1,sw}$ = 11.4 W.

As for the (pre)regulator, it is a DC/DC chopper operating with a DC-link voltage on one port and battery voltage on the other. A versatile solution is a buck–boost scheme, which is also comparable with the simplest PPC realization based on the flyback. One port of this chopper handles the DC-link voltage (the amplitude of the grid voltage $V_{s,m}$) and

average current, corresponding to the power transmitted to/from the grid $P_s$, i.e., $V_s I_s / V_{s,m}$ = $I_s/1.41 = I_{s,m}/2$. This current is delivered with pulses, the value of which can be found in the voltage transfer equation of the buck–boost converter, which, in discharge mode, is

$$V_{s,m} = V_{bat} \frac{D_{bat}}{(1 - D_{bat})} \text{ that gives } D_{bat} = \frac{V_{s,m}}{V_{bat} + V_{s,m}} \tag{16}$$

And the power balance of the converter is as follows:

$$V_{s,m} I_{DC} = V_{bat} I_{bat} \tag{17}$$

where $D_{bat}$ is relative to the time of the switch at the battery side. It is also assumed that the fully charged battery produces a voltage slightly lower than the grid voltage amplitude $V_{s,m}$ (with number of cells $N_{cell} = 75$) and the current in the inductor of the chopper has low ripples (operation in continuous current mode).

Then, the value of the current pulses in the switches, as well as the current of the inductor can be expressed as

$$I_{La} = \frac{I_{s,m}}{2} \frac{1}{1 - D_{bat}} = \frac{I_{s,m}}{2} \frac{V_{bat} + V_{s,m}}{V_{bat}} \tag{18}$$

Since, in the discontinuous conduction mode, one of the switches conducts this current, the conduction losses can be calculated as

$$\Delta P_{REG1,c} = V_0 I_{La} \tag{19}$$

Applying the commutated current $I_{sw} = I_{La}$ and commutated voltage $V_{sw} = V_{s,m} + V_{bat}$ in (12),

$$\Delta P_{REG1,sw} = (V_{s,m} + V_{bat}) I_{La} \left( \frac{t_i}{T_{sw}} + \frac{t_v}{T_{sw}} \right) \tag{20}$$

and it assumed that these parameters affect the corresponding current/voltage changes as follows:

$$\frac{t_i}{T_{sw}} = \frac{t_{i,m}}{T_{sw}} \frac{I_{La}}{I_{s,m}} = \frac{RDS}{2} \frac{I_{La}}{I_{s,m}} \text{ and } \frac{t_v}{T_{sw}} = \frac{t_{v,m}}{T_{sw}} \frac{V_{s,m} + V_{bat}}{V_{s,m}} = \frac{RDS}{2} \frac{V_{s,m} + V_{bat}}{V_{s,m}} \tag{21}$$

It becomes possible to calculate the corresponding switching losses of the regulator and the total losses of the reference converter (Table 3). The losses of the regulator are more than twice as high as those of the frontend—mostly due to the almost doubled commutated voltage of the buck–boost regulator. The power losses of other regulators are potentially lower.

**Table 3.** Power losses in case of full power switch mode regulator and inverter.

| SOC [%] | 0 | 13 | 25 | 38 | 50 | 63 | 75 | 88 | 100 |
|---|---|---|---|---|---|---|---|---|---|
| $V_{bat}$, [V] | 188 | 203 | 219 | 235 | 251 | 267 | 283 | 299 | 315 |
| $V_{sw} = V_{s,m} + V_{bat}$, [V] | 513 | 529 | 545 | 561 | 577 | 592 | 608 | 624 | 640 |
| $I_{sw} = I_{La}$, [A] | 8.41 | 7.99 | 7.63 | 7.32 | 7.05 | 6.82 | 6.61 | 6.42 | 6.25 |
| $t_i/T_{sw}$ [%] | 0.68 | 0.65 | 0.62 | 0.60 | 0.57 | 0.55 | 0.54 | 0.52 | 0.51 |
| $t_v/T_{sw}$ [%] | 0.29 | 0.31 | 0.34 | 0.36 | 0.39 | 0.41 | 0.44 | 0.46 | 0.48 |
| $\Delta P_{REG1,c}$ [W] | 16.8 | 16.0 | 15.3 | 14.6 | 14.1 | 13.6 | 13.2 | 12.8 | 12.5 |
| $\Delta P_{REG1,sw}$ [W] | 41.9 | 40.7 | 39.8 | 39.3 | 39.0 | 39.0 | 39.1 | 39.3 | 39.7 |
| $\Delta P_1$ [W] | 77.9 | 75.8 | 74.3 | 73.1 | 72.3 | 71.8 | 71.5 | 71.4 | 71.4 |
| $\Delta P_1$ [%] | 7.8 | 7.6 | 7.4 | 7.3 | 7.2 | 7.2 | 7.1 | 7.1 | 7.1 |

### 5.2.3. Losses in Case of Partial Power Regulator

Let us now consider the losses of the BESS interface with a partial power (pre)regulator. The applied converter topology is flyback—a version of buck–boost equipped with split

coil (transformer). Therefore, its static voltage equation remains, but, due to the series connection at the DC-link side, can be rewritten as

$$V_{s,m} - V_{bat} = V_{bat} \frac{D_{bat}}{(1 - D_{bat})} \tag{22}$$

which produces

$$D_{bat} = \frac{V_{s,m} - V_{bat}}{V_{s,m}} \tag{23}$$

With this configuration, the commutated voltage (the sum of the input and output voltages of the converter) is fixed at the level $V_{DC} = V_{s,m}$, i.e., the lower the battery voltage the larger the part that is added by the DC-DC converter. In turn, the commutated current can still be expressed by (20). However, in this configuration, it does not flow explicitly in an inductor but is formed as the sum of two currents of primary and secondary windings that in explicit form can be measured in the battery.

As for the frontend, since the schematic and operation remain unchanged, its losses also remain on the previously calculated level of $\Delta P_{FE2,c}$ = 7.8 W and $\Delta P_{FE2,sw}$ = 11.4 W. The power losses corresponding to the partial power regulator are given in Table 4. The positive effect of the operation with reduced voltage and current is clear, even in the case of the considered flyback converter.

**Table 4.** Power losses in case of switch mode inverter and partial power regulator.

| SOC [%] | 0 | 13 | 25 | 38 | 50 | 63 | 75 | 88 | 100 |
|---|---|---|---|---|---|---|---|---|---|
| $V_{bat}$, [V] | 188 | 203 | 219 | 235 | 251 | 267 | 283 | 299 | 315 |
| $V_{sw} = V_{s,m}$, [V] | 325 | 325 | 325 | 325 | 325 | 325 | 325 | 325 | 325 |
| $I_{sw} = I_{bat}$, [A] | 5.33 | 4.92 | 4.56 | 4.25 | 3.98 | 3.74 | 3.53 | 3.34 | 3.17 |
| $t_i/T_{sw}$ [%] | 0.43 | 0.40 | 0.37 | 0.35 | 0.32 | 0.30 | 0.29 | 0.27 | 0.26 |
| $t_v/T_{sw}$ [%] | 0.29 | 0.31 | 0.34 | 0.36 | 0.39 | 0.41 | 0.44 | 0.46 | 0.48 |
| $\Delta P_{REG2,c}$ [W] | 10.7 | 9.8 | 9.1 | 8.5 | 8.0 | 7.5 | 7.1 | 6.7 | 6.3 |
| $\Delta P_{REG2,sw}$ [W] | 12.5 | 11.4 | 10.5 | 9.8 | 9.2 | 8.7 | 8.3 | 8.0 | 7.7 |
| $\Delta P_2$ [W] | 42.39 | 40.42 | 38.81 | 37.47 | 36.35 | 35.39 | 34.56 | 33.84 | 33.21 |
| $\Delta P_2$ [%] | 4.2 | 4.0 | 3.9 | 3.7 | 3.6 | 3.5 | 3.5 | 3.4 | 3.3 |

### 5.2.4. Losses in Case of Unfolding Frontend

In the case of the unfolding frontend, the grid current in each half-period constantly flows through the couple of the frontend switches. For this reason, the frontend produces only conduction losses that can still be calculated with (9) as $\Delta P_{FE3,c}$ = 7.8 W while $\Delta P_{FE3,sw}$ = 0 W.

For such a frontend, the (pre)regulator forms semi-sine voltage half-waves in the "virtual" DC-link (which includes a small capacitor, capable of reducing only high-frequency voltage ripples). This is why most of the basic parameters are functions of the grid phase:

$$\text{voltage } V_{s,k} = V_{s,m} \sin\theta_k$$

$$\text{and current in virtual DC} - \text{link } I_{sw,k} = I_{s,k} = I_{s,m} \sin\theta_k \tag{24}$$

commutated voltage

$$V_{sw,k} = V_{bat} + V_{s,k} = V_{bat} + V_{s,m} \sin\theta_k \tag{25}$$

$$\text{static equation } V_{s,m} \sin(\theta_k) = V_{bat} \frac{D_{bat,k}}{1 - D_{bat,k}} \tag{26}$$

$$\text{and duty cycle } D_{bat,k} = \frac{V_{s,m} \sin\theta_k}{V_{bat} + V_{s,m} \sin\theta_k} \tag{27}$$

Other parameters, therefore, are also expressed as such functions. The commutated (inductor) current can be found in the DC-link capacitor balance which requires

$$I_{La,k}(1 - D_{bat,k}) = I_{s,k} \text{ giving } I_{sw,k} = I_{La,k} = \frac{I_{s,m}\sin\theta_k}{1 - D_{bat,k}} \tag{28}$$

The conduction losses, dissipated in two switches (one of which is conducting), can then be expressed as

$$\Delta P_{REG3,c,k} = \frac{V_0 I_{s,k}}{1 - D_{bat,k}} \tag{29}$$

but applying (27)¯as $\Delta P_{REG3,c,k} = \dfrac{V_{s,k}V_0 I_{s,k}}{V_{bat}} + V_0 I_{s,k}$

$$\Delta P_{REG3,c,k} = \frac{V_{s,m}V_0 I_{s,m}\sin_2\theta_k}{V_{bat}} + V_0 I_{s,m}\sin\theta_k \tag{30}$$

Applying (29) into the equation of averaged (active) power

$$\Delta P_{REG3,c} = \frac{1}{0.5 T_s}\sum_{k=1}^{N}\Delta P_{REG3,c,k}T_{sw} = \frac{1}{\pi}\int_{0}^{\pi}\Delta P_{REG3,c}(\theta)d\theta \tag{31}$$

after simplifications produces

$$\Delta P_{REG3,c} = \frac{1}{2}\frac{V_0}{V_{bat}}V_{s,m}I_{s,m} + \frac{2}{\pi}\frac{V_0}{V_{s,m}}V_{s,m}I_{s,m} = \frac{V_0}{V_{bat}}P_s + \frac{4}{\pi}\frac{V_0}{V_{s,m}}P_s \tag{32}$$

The switching losses of the pulse mode regulator can still be calculated by (12) utilizing the commutated current (24) and commutated voltage (25). The analytical solution of the corresponding formula refers to a sine-form signal in power 3 or even 4. For this reason, the switching losses are calculated numerically in general form. Then, (12) can be rewritten as

$$\Delta P_{REG3,sw,k} = \frac{1}{T_{sw}}\cdot 2\cdot \frac{V_{sw,k}\cdot I_{sw,k}\cdot(t_{i,k}+t_{v,k})}{2} = \frac{1}{T_{sw}}\cdot 2\cdot \frac{(V_{bat}+V_{s,k})\frac{I_{s,k}}{1-D_{bat,k}}\cdot(t_{i,k}+t_{v,k})}{2}$$

$$\text{with } \frac{t_{i,k}}{T_{sw}} = \frac{t_{i,m}}{T_{sw}}\frac{I_{La,k}}{I_{s,m}} = \frac{RDS}{2}\frac{I_{s,k}}{1-D_{bat,k}}\frac{1}{I_{s,m}} \tag{33}$$

$$\text{and } \frac{t_{v,k}}{T_{sw}} = \frac{t_{v,m}}{T_{sw}}\frac{V_{bat}+V_{s,k}}{V_{s,m}} = \frac{RDS}{2}\frac{V_{bat}+V_{s,k}}{V_{s,m}}$$

The results of the conduction loss calculation with (29), their verification with (32) and switching loss calculation with (33) are presented in Table 5. The number of switching cycles in these calculations is 100 which corresponds to a switching frequency at 10 kHz. It is seen that the general level of the losses is comparable with the reference design (pulse mode frontend, pulse mode full power regulator), but some parts of the switching losses "moved" from the frontend to the regulator.

5.2.5. Losses in Case of Unfolding Inverter and Partial Power Regulator

Let us apply the above-scribed loss calculation procedure to the proposed battery interface still assuming that the DC-DC converter is a classical inverting buck–boost chopper.

Like in the previous case the frontend is an unfolding inverter, where the grid current in each half-period constantly flows through two switches and the frontend losses are purely conduction losses calculated with (9) as $\Delta P_{FE4} = \Delta P_{FE4,c} = 7.8$ W.

In addition, the proposed and evaluated configuration utilizes an extra unfolding inverter (Figures 1b,d and 9). Like a grid inverter, this unfolder conducts semi-sine half-waves (although the commutations occur not only at grid phases 0° and 180°). This

means that at the applied assumptions, this inverter produces the same conduction losses $\Delta P_{UF+} = \Delta P_{UF+,c} = 7.8$ W (without significant switching losses).

**Table 5.** Power losses in case of unfolding inverter and full power regulator.

| SOC [%] | 0 | 13 | 25 | 38 | 50 | 63 | 75 | 88 | 100 |
|---|---|---|---|---|---|---|---|---|---|
| $V_{bat}$, [V] | 188 | 203 | 219 | 235 | 251 | 267 | 283 | 299 | 315 |
| $V_{sw,\,max}$, [V] | 513 | 529 | 545 | 561 | 576 | 592 | 608 | 624 | 640 |
| $V_{sw,\,min}$, [V] | 193 | 209 | 224 | 240 | 256 | 272 | 288 | 304 | 320 |
| $D_{max}$ [%] | 63 | 62 | 60 | 58 | 56 | 55 | 53 | 52 | 51 |
| $D_{min}$ [%] | 3 | 2 | 2 | 2 | 2 | 2 | 2 | 2 | 2 |
| $\Delta P_{REG3,c,\,discrete}$ [W] | 18.5 | 17.7 | 16.9 | 16.3 | 15.8 | 15.3 | 14.9 | 14.5 | 14.2 |
| $\Delta P_{REG3,c,\,exact}$ [W] | 18.5 | 17.7 | 16.9 | 16.3 | 15.8 | 15.3 | 14.9 | 14.5 | 14.2 |
| $\Delta P_{REG3,sw}$ [W] | 75.3 | 73.1 | 71.5 | 70.5 | 69.8 | 69.5 | 69.4 | 69.5 | 69.9 |
| $\Delta P_3$ [W] | 101.6 | 98.6 | 96.3 | 94.6 | 93.4 | 92.6 | 92.1 | 91.9 | 91.9 |
| $\Delta P_3$ [%] | 10.2 | 9.9 | 9.6 | 9.5 | 9.3 | 9.3 | 9.2 | 9.2 | 9.2 |

The main considerations applied to the calculation of losses of the regulator, connected to the unfolding inverter, can also be expanded to this configuration with some corrections.

In static Equation (26), the rectified grid voltage $V_{s,k}$ has to be changed to the rectified difference of $V_{s,k}$ and battery voltage $V_{s,k}$ (as in Figures 6b–8b). Then, the static equation and expression for the duty cycle calculation is as follows:

$$\left| V_{s,k} - V_{bat} \right| = V_{bat} \frac{D_{bat,k}}{1 - D_{bat,k}} \tag{34}$$

$$\text{and } D_{bat,k} = \frac{\left| V_{s,k} - V_{bat} \right|}{V_{bat} + \left| V_{s,k} - V_{bat} \right|},$$

$$\text{which can be } D_{bat,k} = \frac{V_{s,k} - V_{bat}}{V_{s,k}} \text{ or } D_{bat,k} = \frac{V_{bat} - V_{s,k}}{2V_{bat} - V_{s,k}} \tag{35}$$

depending on the grid phase.

The static current balance (28) for determining the commutated current remains, but has to be composed for internal capacitor C21 and utilized (35) for the duty cycle.

Finally, the commutated voltage also refers to the rectified difference of $V_{s,k}$ and battery voltage $V_{s,k}$

$$V_{sw,k} = V_{bat} + \left| V_{s,k} - V_{bat} \right| \text{that can be } V_{sw,k} = V_{s,k} \text{ or } V_{sw,k} = 2V_{bat} - V_{s,k} \tag{36}$$

The analytical calculation of the losses requires applying (28), (35), (36) to (29) and (33) and a further complicated integration. These equations, therefore, have been used numerically. The calculations for a 64 cells battery, providing a voltage of 160–270 V are presented in Table 6.

**Table 6.** Power losses in case of unfolding inverter and partial power regulator (battery is composed of 64 cells).

| SOC [%] | 0 | 13 | 25 | 38 | 50 | 63 | 75 | 88 | 100 |
|---|---|---|---|---|---|---|---|---|---|
| $V_{bat}$, [V] | 160 | 174 | 187 | 201 | 214 | 228 | 242 | 255 | 269 |
| $V_{sw,\,max}$, [V] | 325 | 342 | 369 | 396 | 424 | 451 | 478 | 505 | 532 |
| $V_{sw,\,min}$, [V] | 161 | 177 | 187 | 203 | 218 | 230 | 243 | 257 | 272 |
| $D_{max}$ [%] | 51 | 49 | 49 | 49 | 49 | 49 | 49 | 49 | 50 |
| $D_{min}$ [%] | 1 | 2 | 0 | 1 | 1 | 1 | 0 | 1 | 1 |
| $\Delta P_{REG4,c}$ [W] | 13.2 | 12.3 | 11.6 | 11.0 | 10.5 | 10.2 | 9.9 | 9.6 | 9.5 |
| $\Delta P_{REG4,sw}$ [W] | 22.3 | 19.8 | 18.0 | 16.6 | 15.6 | 15.0 | 14.7 | 14.7 | 15.1 |
| $\Delta P_4$ [W] | 51.1 | 47.8 | 45.2 | 43.3 | 41.8 | 40.8 | 40.2 | 40.0 | 40.2 |
| $\Delta P_4$ [%] | 5.1 | 4.8 | 4.5 | 4.3 | 4.2 | 4.1 | 4.0 | 4.0 | 4.0 |

Additional calculations (Tables 7 and 8) have also been made for batteries with 50 and 75 cells. These calculations in conjunction with the previously obtained data are graphically compared in Figure 13. This diagram proves that the considered BESS interface still has the advantages of partial power converters though they are not so visible as in the case of DC systems. The graphics also show that the highest efficiency can be achieved at a battery voltage of about 255 V (at which there is no circulating power flow through the DC-DC regulator). Therefore, the battery with this voltage in the middle of its operation range can be considered as optimal.

**Table 7.** Power losses in case of unfolding inverter and partial power regulator (battery made of 50 cells).

| SOC [%] | 0 | 13 | 25 | 38 | 50 | 63 | 75 | 88 | 100 |
|---|---|---|---|---|---|---|---|---|---|
| $V_{bat}$, [V] | 125 | 136 | 146 | 157 | 168 | 178 | 189 | 199 | 210 |
| $\Delta P_4$ [%] | 6.5 | 6.0 | 5.5 | 5.2 | 4.9 | 4.7 | 4.5 | 4.3 | 4.2 |

**Table 8.** Power losses in case of unfolding inverter and partial power regulator (battery made of 75 cells).

| SOC [%] | 0 | 13 | 25 | 38 | 50 | 63 | 75 | 88 | 100 |
|---|---|---|---|---|---|---|---|---|---|
| $V_{bat}$, [V] | 188 | 203 | 219 | 235 | 251 | 267 | 283 | 299 | 315 |
| $\Delta P_4$ [%] | 4.5 | 4.3 | 4.1 | 4.0 | 4.0 | 4.0 | 4.1 | 4.2 | 4.4 |

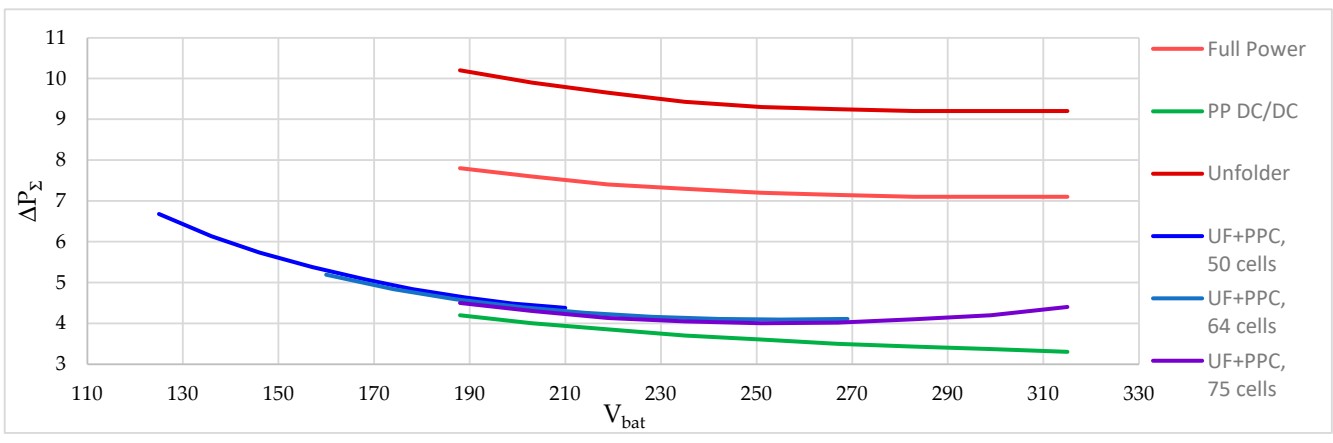

**Figure 13.** Comparison of power losses in the considered systems.

5.2.6. General Considerations on Calculation of Converter Losses

The battery interface converter with a flyback pre-regulator, considered in Section 5.2, is derived from the inverting buck–boost chopper that calculates the sum of its input and output voltages. So, the flyback converter on the primary side calculates the sum of the primary voltage and "reflected" secondary voltage, but on the secondary side, the sum of the secondary voltage and "reflected" primary voltage is obtained. The unity ratio of turns for the primary and secondary windings of the transformer on both ends converter's switches commutate the sum of voltages. In the case of BESS, one of these voltages is the battery voltage, but the other is the full or partial DC-link voltage. Therefore, the commutated voltage directly depends on the battery.

In turn, the commutated voltage, with the selected loss calculation methodology has a direct influence on the switching losses in the pre-regulator; this voltage is placed in the numerator of the corresponding formula. It has also an indirect influence on these losses because this voltage defines the voltage rise/fall times. For this reason, the switching losses with such a configuration are unreasonably higher compared to the case of other topologies of the pre-regulator that typically calculate the maximal input/output voltage. This voltage

overrating can be evaluated as follows: (190–315 + 315)/315 = 1.6–2 for the full-power full-switching interface converter and interface with an unfolder, 315/(190–315) = 1–1.7 for the partial-power full-switching BESS interface and (190–315 + 315/2)/190–315 = 1.5–1.8 for the proposed converter.

## 6. Conclusions

After the experimental verification and analysis of the mathematical model for the loss calculation, the proposed battery interface converter that combines an unfolding inverter as the grid frontend with a partial power DC-DC converter as (pre)regulator was found operational. A more detailed explanation of this matter includes the following considerations:

(1) The described series configuration of the utilized battery and output port of the applied DC-DC converter is capable of generating the semi-sine voltage halfwaves that can be transformed by unfolding the inverter into a sine voltage at the grid port of the interface converter. This consideration is valid for multiple configurations of the battery.

(2) The proposed configuration keeps the advantage of the BESS interface with an unfolding inverter—the absence of bulky and less reliable DC-link electrolytic capacitors, calculated for grid frequency.

(3) From an efficiency point of view, the proposed configuration behaves as PPC. Its losses are lower than those of the reference full-power converters, but not as low as the losses in the case of a DC-DC PPC. This can be explained by a larger voltage, to be compensated by the PPC (pre)regulator, as well as by the dynamic nature of this compensation—even at no average power transfer through PPC (at 255 V), there is always some instantaneous power through the regulator that leads to losses. From this point of view, preferences should be given to the batteries with voltages around 255 V (assuming grid voltage of 230 $V_{AC}$) and battery chemistries that provide lower voltage difference vs. SOC. The level of the losses obtained in this work is rather high. However, it is obtained for inverting buck–boost and flyback regulators, which are more convenient for comparison, but calculate the sum of input and output voltages which leads to higher switching losses (with the considered model).

(4) From the point of view of the current ratings of the switches of the proposed configuration, it is not very advantageous. There are instances in time when the regulator conducts a full grid current. On the other hand, the commutated voltage depends on the battery. At the lowest level, the battery and regulator voltages are equal and are twice as low as the grid voltage amplitude.

The above-mentioned considerations allow us to conclude that the proposed BESS interface keeps the major advantages of the applied loss minimization techniques (PPC and UF). On one hand, compared to the full-power BESS interface with the unfolding inverters, the proposed interface converter provides lower conduction and commutation losses in the voltage pre-regulator (in addition to its smaller size, lower weight and higher reliability due to a twice lower voltage stress on the switches). On the other hand, compared to the switching mode BESS interface with a partial power regulator, the proposed converter provides lower losses in the frontend stage (in addition to the higher reliability of the entire interface converter due to the absence of high-capacity electrolytic capacitors).

Further improvement of this research may include the experimental study of the converter with real grid and multiple configurations of the battery, an experimental evaluation of its efficiency and losses in various operation modes and considering the losses in passive components, as well as the selection of the optimal battery chemistry and its configuration for the proposed battery interface. Particular attention should be paid to the improvement of the proposed and studied BESS interface converter by means of the better choice of the pre-regulator. As has been mentioned, the flyback regulator is not optimal from the point of view of the commutated voltage and switching losses and an alternative has to be chosen. The choice, already mentioned in this paper—DAB with an extra unfolder—due to the

doubled number of transistors in current paths, seems imperfect from the point of view of the conduction losses.

Another potential option—a bidirectional and bipolar push-pull converter—looks more promising. Another way of improvement is splitting the bidirectional regulator into battery charging and battery discharging parts (Figure 1c,d). This could potentially reduce the number of conducting switches and the corresponding switching losses. However, its use may make the trade-off between converter cost and losses more significant. Finally, the performance of the converter can be improved by the reasonable combining of traditional Si semiconductor devices with wide bandgap (SiC and GaN) switches. The proposed BESS interface may have quite distinct allocations of the switching and conduction losses and, therefore, the wise use of SiC and GaN switches looks promising.

The last, but not least, consideration is the influence of the proposed BESS interface converter on the operation parameters of the battery: temperature, state of health and overall lifetime. From this point of view, the most significant factor is the shape of battery charge/discharge currents. It can be guessed that independently of the functional structure (Figure 1) and with particular implementation (Figures 2–5, 9 or other), this current can be constructed as the sum of the currents of the DC/DC converter at its DC-link and battery ports. The first one (Figure 11d) has a semi-sine shape, but the second one can be derived from the instantaneous power of the DC-DC converter (Figure 12) taking into account the DC voltage that finally gives the current at the battery port, shown in Figure 11a. It is clear that this current is not a DC current, comfortable for batteries, but at the same time, it is not a pulse mode current—the most difficult for them. The current is constructed of several sine pieces. The influence of such a current on the battery is not clear and its determination requires multidisciplinary (electrical, chemical and heat exchange) research, dedicated to this topic. Such research is also planned as future work.

**Author Contributions:** Conceptualization, I.A.G. and A.B. (Andrei Blinov); methodology, I.A.G. and R.S.; validation, R.S, A.B. (Alexander Bubovich) and I.A.G.; formal analysis, D.P.; investigation, I.A.G.; writing—original draft preparation, I.A.G.; writing—review and editing, A.B. (Alexander Bubovich) and I.A.G.; visualization, I.A.G.; supervision, I.A.G.; project administration, A.B. (Andrei Blinov); funding acquisition, A.B. (Andrei Blinov). All authors have read and agreed to the published version of the manuscript.

**Funding:** This work was supported in part by the European Economic Area (EEA) and Norway Financial Mechanism 2014–2021 under grant EMP474 "Optimized Residential Energy Storage Systems". This work was supported in part by the European Regional Development Fund (ERDF) under contract 1.1.1.1/20/A/079 "Research and Development of Two-Phase Thermal Systems Installed in Lighting Equipment for its Functional Improvement" and contract Nr. 1.1.1.1/16/A/147 "Research and Development of Electrical, Information and Material Technologies for Low Speed Rehabilitation Vehicles for Disabled People" of its Latvian measure 1.1.1.1 "Industry-Driven Research".

**Data Availability Statement:** Data are contained within the article.

**Conflicts of Interest:** The authors declare no conflicts of interest.

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
