# Peer review of "Considerations on Combining Unfolding Inverters with Partial Power Regulators in Battery–Grid Interface Converters"

_energies, doi:10.3390/en17040893_

Round 1

Reviewer 1 Report

Comments and Suggestions for Authors

The paper presents a study on combining unfolding inverters with 2 partial power regulators for Battery application. The study includes experimental verification, Here are my comments on the paper:

1.       Please improve this sentence “ The aim of this work is to achieve further loss reduction in the mentioned rather high voltage BESS not only due to their operation at lower currents”  for better readability.

2.       What distinguishes isolated converters from non-isolated converters in Battery Energy Storage Systems (BESS), and what are the safety considerations associated with each type?

3.       How do non-isolated converters, in particular, contribute to potentially higher energy efficiency in BESS, and what trade-offs are involved compared to isolated converters?

4.       What is the significance of introducing a pre-regulator in single-stage converters, and how does it impact the efficiency and operation range?

5.       How does the commutated voltage in the proposed configuration depend on the battery, and what are the implications of this dependence?

6.       What potential areas for further improvement are identified in the research, and what specific aspects could be addressed in an experimental study of the converter with real grid and various battery configurations?

7.       What advantage does the proposed configuration maintain in terms of the BESS interface with the unfolding inverter, particularly in comparison to full-power converters?

Comments on the Quality of English Language

Moderate editing of English language is required

Author Response

Thank you very much for your remarks, questions and recommendations. The corresponding answers can be found in the attached file.

Sincerely,

Authors. 

Reviewer 2 Report

Comments and Suggestions for Authors

This article is related to electronic converters applied to charging and discharging batteries and, more specifically, to the study of a topology for minimizing losses in the DC-DC stage.

From the summary and the introduction, it is difficult to know what is new about the article. The authors refer to other published works in these manuscript sections without detailing their achievements.

It would be necessary for the authors to introduce changes in the summary and introduction, including the methodology used to avoid the problem and the main novelties achieved.

Specifically, the authors should indicate in the Introduction whether they have developed a new converter different from those known in the Technical Literature or whether they have used it differently and, in the latter case, highlight the differences in the results.

On the other hand, the authors focus their study on reducing losses during battery charging and discharging. However, the authors have analyzed what effects the ripple of the converter output voltages has on the heating and useful life of the batteries.

The bibliography used is adequate.

Differences in the transformer symbol in Table 1 (T12) and Figure 9 (T1) should be corrected.

Comments on the Quality of English Language

Minor editing of the English language is required.

Author Response

(The authors gave the same response as above.)

Round 2

Reviewer 2 Report

Comments and Suggestions for Authors

The changes made to the manuscript are appropriate, and I believe the article can be published.

Author Response

Dear Colleague,

Thank you very much for considering of our paper one again, as well as for positive decision.

Sincerely,

Authors.